# Antimicrobial and the Resistances in the Environment: Ecological and Health Risks, Influencing Factors, and Mitigation Strategies

**DOI:** 10.3390/toxics11020185

**Published:** 2023-02-16

**Authors:** Weitao Wang, You Weng, Ting Luo, Qiang Wang, Guiling Yang, Yuanxiang Jin

**Affiliations:** 1College of Biotechnology and Bioengineering, Zhejiang University of Technology, Hangzhou 310032, China; 2State Key Laboratory for Managing Biotic and Chemical Threats to the Quality and Safety of Agro-Products, Laboratory (Hangzhou) for Risk Assessment of Agricultural Products of Ministry of Agriculture, Institute of Agro-Product Safety and Nutrition, Zhejiang Academy of Agricultural Sciences, Hangzhou 310021, China

**Keywords:** antimicrobial, antimicrobial resistance, environment factors, human health, mitigation methods

## Abstract

Antimicrobial contamination and antimicrobial resistance have become global environmental and health problems. A large number of antimicrobials are used in medical and animal husbandry, leading to the continuous release of residual antimicrobials into the environment. It not only causes ecological harm, but also promotes the occurrence and spread of antimicrobial resistance. The role of environmental factors in antimicrobial contamination and the spread of antimicrobial resistance is often overlooked. There are a large number of antimicrobial-resistant bacteria and antimicrobial resistance genes in human beings, which increases the likelihood that pathogenic bacteria acquire resistance, and also adds opportunities for human contact with antimicrobial-resistant pathogens. In this paper, we review the fate of antimicrobials and antimicrobial resistance in the environment, including the occurrence, spread, and impact on ecological and human health. More importantly, this review emphasizes a number of environmental factors that can exacerbate antimicrobial contamination and the spread of antimicrobial resistance. In the future, the timely removal of antimicrobials and antimicrobial resistance genes in the environment will be more effective in alleviating antimicrobial contamination and antimicrobial resistance.

## 1. Introduction

Antimicrobials, which are effective in combatting pathogenic microorganisms, have been widely used in the prevention and treatment of diseases and livestock farming since their discovery in the 20th Century [1,2,3]. Millions of lives have been saved by the advent of antimicrobials, of which more than 100,000 tons per year are used worldwide [4]. At the same time, the extensive use of antimicrobials has inevitably led to leakage and residue in the environment [5]. Antimicrobials are discharged through host metabolism or directly released into nature, especially in aquaculture and agricultural applications [6]. Although the half-life of antimicrobials is relatively short, often ranging from a few hours to dozens of days, human abuse and continuous release have led to the detection of various antimicrobials and their decomposition products in wastewater, groundwater, and surface water [7,8]. In addition, residues of antimicrobials have been detected in milk, eggs, vegetables, and other foods, which cause long-term exposure to antimicrobials [9]. However, some studies have reported that antimicrobial residues can cause certain toxic effects on organisms [1,10,11]. Consequently, antimicrobial contamination has become a global problem [12].

Antimicrobials are considered to be a crucial driving factor in the generation and spread of antimicrobial-resistant bacteria (ARB) and antimicrobial resistance genes (ARGs) [13,14]. Microorganisms, especially clinical pathogens, acquire ARGs via horizontal gene transfer (HGT) from the environment, which reduces sensitivity to antimicrobials [13,15]. Microbes rapidly multiply and can freely spread in the environment, facilitating the transmission of antimicrobial resistance. ARGs have accelerated microbial threats to human health over the past decade [16]. In 2019, antimicrobial-resistant infections were directly or indirectly responsible for an estimated 6.22 million deaths worldwide [17]. Antimicrobial resistance has already become a considerable global public health security issue [18]. This is due to the limited ability of wastewater treatment plants to remove antimicrobials, ARGs, and ARB, which are widely regarded as emerging environmental pollutants [19,20,21]. Antimicrobials, ARB, and ARGs have been found in a variety of environments [22]. In addition, ARGs have been detected in atmospheric environments and PM2.5, which increases the exposure risk to humans [23,24].

In the human body, the gut is the main organ for digestion and absorption, as well as the main harbor of microorganisms. Recently, we have come to appreciate the importance of gut microbiota in human health [25,26,27]. Disorders of the gut microbiota may lead to diseases; for example, obesity and chronic kidney disease [28,29]. However, the risk of antimicrobials and ARGs to humans is principally caused by their acting on gut microbes [30]. HGT is more likely to occur in places like the gut that contain abundant microbes [31]. The intake of food or water with residual antimicrobials and ARGs is terrible, which reduces the bacterial diversity of gut microbes and increases ARB colonization and ARGs amplification, thus lowering resistance to pathogen intrusion [32]. Therefore, an insight into the risks of antimicrobials and ARGs to the ecological environment and human health is essential.

Nature is a complex environment where antimicrobials interact with other factors or pollutants, which pose a greater threat to ecosystems and human health; in addition to the various pollutants produced by human activities [33,34]. These pollutants have influence on the enrichment and transfer of ARGs and the generation of multi-drug-resistant bacteria [35,36]. These variable and uncontrollable factors may lead to more serious harm from ARGs and antimicrobials.

In this study, we summarized the relationship between antimicrobials, ARB, and ARGs, residues in the environment, and existing removal methods based on the latest literature. In addition, this study analyzed various factors concerning some contaminants in the complex environment affecting the transfer and enrichment of antimicrobial resistance and ARGs. We also discussed the risks and threats of antimicrobials and antimicrobial resistance to the ecological environment and human health. Finally, we hope to attract more attention to and research on the removal of antimicrobials and ARGs to reduce the negative effects of antimicrobial resistance through this review.

## 2. The Usage and Fate of Antimicrobials in the Environment

Antimicrobials are a class of compounds with killing and inhibiting effects on bacteria, most of which are naturally synthesized [37]. The development of antimicrobials was considered one of the most important medical breakthroughs of the 20th Century [3]. Since the discovery of penicillin, antimicrobials have enjoyed a golden age of development. A variety of antimicrobials have been discovered and applied to the prevention of disease, and have become an important part of the field of modern medical care [38]. Today, more than 150 antimicrobials are currently in use, which according to the chemical structure, can be divided into aminoglycoside antimicrobials, quinolone antimicrobials, macrolide antimicrobials, β-lactam antimicrobials, etc. [39]. The mechanisms of the bactericidal action of antimicrobials are: (1) to inhibit bacterial cell wall synthesis; (2) interact with the cell membrane to change membrane permeability; (3) to interfere with protein synthesis; (4) to inhibit the replication and transcription of nucleic acid [3]. In addition to clinical applications, antimicrobials have also been widely used in agriculture to control pests and diseases, and as feed additives to promote the growth of livestock [2,40]. In fact, the medical use of antimicrobials in China and the United States tends to be smaller than in livestock, agriculture, and aquaculture [41]. In order to meet the needs of medical treatment and production, the mass production and use of antimicrobials has become a global trend. Fortunately, people have begun to realize that the abuse and overuse of antimicrobials pose a threat to human health and the environment, and have begun to advocate reducing the unnecessary use of antimicrobials and seeking safer alternatives to antimicrobials.

### 2.1. Entry of Antimicrobials into the Environment

Antimicrobials enter the environment in a variety of ways and affect human health and ecology (Figure 1). Water and soil are the main repositories for antimicrobials. Due to their low bioavailability, after being metabolized by pigs, the concentrations of sulfonamides in feces and urine were 15.03–26.55% and 4.54–69.22%, respectively [42]. A similar study showed that the macrolide drug Solimycin was 76.5% and 14.1% in feces and urine, respectively, when orally taken by patients [43]. A study by Chen et al. reported that the fate of antimicrobial emissions in China mainly concentrates in human and livestock feces, accounting for 57.6% and 42.6% of medicinal antimicrobials and veterinary antimicrobials, respectively [5]. However, on the one hand, the excrement of livestock and poultry has mainly been used as fertilizer for agriculture, and the residual antimicrobials have leached into the soil, reaching groundwater, surface water, and other aquatic environments through runoff [44]. It has been estimated that 5711 tons of antimicrobials were released into the lakes of China in 2019 [45]. On the other hand, human waste enters wastewater treatment plants (WWTPs) through domestic wastewater sewer lines. Nevertheless, WWTPs have reported limited effectiveness in diminishing antimicrobials [46,47]. WWTPs-treated sewage has been discharged into surface water, and activated sludge in WWTPs has been reused as biological fertilizer, which has caused antimicrobials to enter the natural water cycle and to spread to soil [48]. Antimicrobials have been observed in wastewater, surface water, drinking water, groundwater, and soil to some extent [49,50,51,52,53], and the concentration is generally between ng/L and mg/L levels [54].

### 2.2. Effects of Antimicrobials on the Environment and Human Health

Cyanobacteria are widely found in aquatic ecosystems, and cyanobacteria blooms will lead to red tides and blooms. In addition to environmental residues, antimicrobials are sometimes used to remove cyanobacterial blooms. However, it has been reported that antimicrobials at an environmental concentration of 300 ng/L have been proved to promote the growth of cyanobacteria and the formation of blooms [55]. In the same way, Jiang et al. came to similar conclusions, and found low doses of antimicrobials would promote the production of microcystin in some cyanobacteria [56]. Antimicrobials in the environment increase the risk of cyanobacterial blooms and the release of microcystin, which affect the ecological balance and indirectly threatened human and animal health [57,58]. Furthermore, as important primary producers in ecosystems, phytoplankton are sensitive to some antimicrobials [59]. Antimicrobials affect phytoplankton growth and inhibit photosynthesis [60]. Due to the detection of a variety of antimicrobials in the environment, research has shifted from the effects of single antimicrobials to the effects of combined antimicrobial exposure [54].

Fish are another major target of antimicrobials in the aquatic environment, although they are less sensitive to antimicrobials than algae [61]. Zebrafish, as an ideal model organism, have been widely used in toxicological studies [62]. In recent years, there have been many studies on the toxicity of zebrafish exposed to various antimicrobials. Macrolide antimicrobials have been found to adversely affect the heart, liver, and development of zebrafish [63,64]. Acute antimicrobial exposure induces behavior changes and cognitive decline in zebrafish [10]. Long-term exposure to oxytetracycline potentially alters brain thyroid hormone and serotonin homeostasis in zebrafish [65]. In addition to these effects, environmental concentrations of antimicrobials also cause intestinal microbiome disorder and impair the intestinal barrier and function of fish [11,40,66]. Notably, antimicrobials have been reported to affect the growth of zebrafish during pregnancy and the survival of the offspring, and antimicrobial residues have been detected in eggs [67]. At the same time, some adverse effects of antimicrobials have also been found in amphibians and mammals [68,69].

In addition to having negative effects on algae and animals, the increasing presence of antimicrobials in the environment also poses a threat to the normal growth of terrestrial plants. In addition, antimicrobials in wastewater irrigation and manure fertilization can also affect crop seeds and root growth, and have a bioaccumulation effect [70]. During the radicle elongation of Chinese cabbage after germination, tetracycline stress disturbs the mobilization of protein bodies in seed storage reserves [71]. Some laboratory studies have shown similar results, such as the use of chlortetracycline in inhibiting seed germination and seedling growth in Brassica campestris [72], and some antimicrobials affect the seed germination and root development of tomatoes [73]. Meanwhile, a recent study found that irrigating crops with water contaminated with enrofloxacin resulted in reduced soybean yields and the accumulation of antimicrobials [74]. Different types and concentrations of antimicrobials have shown adverse effects on the normal growth of organisms (Table 1). Notably, the harm of antimicrobial pollution is not only manifested in the impact on plant growth and development, but also may cause bioconcentration through the food chain, and thus affect human health.

Actually, anaphylaxis is the most common adverse event of clinical antimicrobial use in humans [75,76]. Vancomycin is an antimicrobial used to treat infections with gram-positive bacteria, and common allergic reactions include itching and rashes [77]. In addition, contemporary guidelines recommend targeting vancomycin trough concentrations of ≥10 mg/L to prevent resistance and trough concentrations of 15–20 mg/L to avoid nephrotoxicity [78]. Therefore, skin pricking (50 mg/mL) and intradermal subcutaneous tests (0.005 mg/mL) at non-irritating concentrations have often been used to determine whether allergies may occur [79]. However, more health risks may come from the effects of antimicrobials on gut microbes. Recently, a growing number of studies have shown that changes in gut microbes are associated with some diseases, such as diabetes, hypertension, and chronic kidney disease [80,81]. A previous study suggested that early antimicrobial exposure may cause childhood asthma, allergic rhinitis, obesity, celiac disease, and other adverse reactions [82]. Dietary ingestion is the other way of human exposure to antimicrobials [5]. Residues of antimicrobials in food can disrupt gut microbes and further damage human health. Remarkably, maternal use of antimicrobials during pregnancy is capable of transmitting to the fetus via the umbilical cord placenta, and antimicrobials have also been detected in breast milk [83]. Because the composition of the gut microbiome in newborns is dominated by the mother and the environment, these antimicrobials delivered to infants will affect normal gut microbiome construction [84]. Consequently, abnormal gut microbiota in infants may lead to reduced resistance to pathogens and an increased risk of some metabolic diseases [85,86]. In addition, antimicrobial resistance is another emerging hotspot and one of the biggest threats facing humanity [87]. The use of antimicrobials has exerted selective pressure on ARB and ARGs [88]. It is especially likely to occur in places where microbes congregate, such as the gut, which is one of the densest bacterial habitats on earth [89]. The rise of ARB and the spread of ARGs has increased the probability of human infection with pathogenic bacteria and reduces the effectiveness of antimicrobial drugs. These results indicate that antimicrobial pollution has certain toxic effects and potential health risks to organisms in the ecological environment.

### 2.3. The Combined Effects of Environmental Pollutants and Antimicrobials

Human activities have produced many pollutants and released them into nature; for example, microplastics (MPs), heavy metals, fluorinated compounds, and pesticides. As an emerging environmental pollutant, MPs have attracted extensive attention and have been widely observed in water [89,90,91,92,93]. Recently, researchers have found that antimicrobials could be adsorbed by these plastic particles with particle sizes of less than 5 microns through hydrogen bonding, hydrophobic interactions, van der Waals forces, and electrostatic interactions [94,95,96]. The adsorption may cause more toxic effects on aquatic organisms. Gonzalez-Pleiter et al. found 50 mg/mL MPs (terephthalate, polylactic acid, polyoxymethylene, and polystyrene) could absorb azithromycin and clarithromycin, which significantly inhibited the growth and chlorophyll content of cyanobacterium. [97]. In addition, it was found clams increased the accumulation of oxytetracycline and flfenicol after the addition of 0.26 mg/L polystyrene microplastics [98,99]. Meanwhile, the immunotoxicity of tetracycin and flufenicol to clams was significantly increased when polystyrene microplastics with a size of 500 nm and a concentration of 26 mg/L were added [100]. More importantly, it can lead to a further enrichment of antimicrobials in humans through the food chain, which has potential health risks.

Heavy metal pollution is another global environmental problem, and various metal ions have been detected in water and soil [101]. Different metal ions can promote or inhibit the adsorption of antimicrobials by microplastics [102]. Recently, a report suggested that interactions between antimicrobials and heavy metals influenced toxic effects; for example, the complexation of Cu^2+^/Cd^2+^ with chlortetracycline produced antagonistic toxicity to cyanobacteria [103]. In addition, some studies have revealed that residues in the environment of tetracycline and sulfonamides could bind metal ions such as Fe^2+^/Cu^2+^ formation of antimicrobial-metal complexes, which had more serious persistence and toxicity [104,105]. Not only that, but the photodegradation of enrofloxacin was impeded by iron complexation [106]. In addition, heavy metal ions can also be adsorbed by microplastics through complexation [107,108]. This implies the convergence of complex pollutants in some environments and might pose greater ecological health risks.

Animal manure is often used as soil fertilizer in agriculture, and residues of antimicrobials in feces may affect the fate of pesticides, such as herbicides [109]. Meanwhile, researchers have discovered that the antimicrobial norfloxacin inhibited the degradation of the herbicide at ambient concentrations in the aquatic environment [12]. Consequently, the coexistence of antimicrobials and pesticides can exacerbate the persistence of pesticides in the environment, thereby increasing the environmental risk of pesticide exposure to organisms. Furthermore, a mixture of amoxicillin, sulfamethoxazole, tetracycline, and ciprofloxacin showed a synergistic effect with herbicide glyphosate on *Microcystis aeruginosa*, and promoted microcystin synthesis and bloom formation [110]. Antimicrobials have been found to enhance the bioavailability of pesticides by disrupting the gut microbiome [111]. These results indicate that the coexistence of various pollutants and antimicrobials in the environment might lead to greater hidden dangers of ecological security.

## 3. The Fate of Antimicrobial Resistance in the Environment

Antimicrobial resistance has become an increasingly serious public health problem, and the primary focus is the occurrence and transmission of ARGs and ARB. Actually, the occurrence of ARB and ARGs is the result of the long-term selection of microorganisms under environmental pressure in nature, and the abuse and misuse of antimicrobials has aggravated this phenomenon [112]. The mechanisms of bacterial resistance to antimicrobials are: (1) to express efflux transport proteins or reduce membrane permeability to prevent antimicrobial entry; (2) to develop enzymes to modify or degrade antimicrobials; (3) to protect or modify antimicrobial targets to lower antimicrobial affinity; (4) to produce proteins that can replace the target function and are not inhibited by antimicrobials [113]. ARB have acquired resistance from ARGs free from the environment or other ARB via HGT, forming multi-drug resistant bacteria [114]. This is undoubtedly a great threat to people and the environment. Recently, ARB and ARGs in nature have been continuously detected, and some mediators in the environment, such as soil and water, have provided stable hosts for ARGs exchange between ARB. Hospital wastewater, aquaculture wastewater, and WWTPs are the main habitats of high ARB and ARGs, and mobile genetic elements (MGEs) may play an unexpectedly significant role in the spread of antimicrobial resistance [115]. ARGs, ARB, and MGEs in the air may have also been often overlooked. According to a report, bioaerosol is a significant route for ARGs spreading in livestock farms, and ARGs in the air have been affected by seasons (such as temperature, humidity, air circulation) [116]. ARB and ARGs in these environments have been continuously enriched through vertical gene transfer or HGT, and have formed multiple antimicrobial-resistant pathogenic bacteria, ultimately penetrating fruits, vegetables, and food included in human daily intake [117,118,119]. Humans are unknowingly surrounded by ARB and ARGs. Remarkably, the prevalence of ARGs has also been driven by animal migration and human activities, such as tourism and the global circulation of foodstuffs [117]. To some extent, humans may be potential carriers of ARGs entering the environment, contributing to the spread of ARGs [120].

### 3.1. The Influence of Various Environmental Factors on Antimicrobial Resistance

#### 3.1.1. Heavy Metals

The factors affecting antimicrobial resistance spread have been reported in many studies, and are similar to the factors affecting antimicrobial contamination mentioned above (Figure 2). It has been reported that residual antimicrobials are the major factor inducing bacteria to accept environmental ARGs and increase the mutation rate of bacteria [21,51]. In addition to antimicrobials, heavy metals also play an important role in the development and spread of antimicrobial resistance. In polluted water sources, ARGs abundance showed a relatively strong correlation with the concentration of heavy metals [121]. Reactive oxygen species are thought to be the inducement of ARGs transfer, but not the only one [122]. The presence of heavy metal ions can cause oxidative stress in bacteria, and then increase intracellular reactive oxygen species levels, which may serve as a potential mechanism of heavy metals promoting the formation of ARB [123]. In addition, studies have revealed that microbial resistance to antimicrobials is related to metal resistance, and there is a cross-protection mechanism in microbes [124]. When bacteria are exposed to heavy metal stress, antimicrobial resistance can be enhanced by increasing the expression of some proteins [125]. Meanwhile, many genes encoding metal and antimicrobial resistance show genetic linkages to some MGEs [126]. Therefore, heavy metals may also play an important role in the spread of antimicrobial resistance, facilitating the acquisition of MGEs and ArGs by bacteria. Moreover, due to the environmental persistence and difficulty in degradation, heavy metals have a longer-term effect on promoting the horizontal transfer of ARGs [127].

#### 3.1.2. MPs

Plastics in the environment provide a new ecological niche for microorganisms, especially MPs, which have gained increasing attention [128]. Microorganisms colonize the surface of MPs, which promotes the exchange and transfer of ARGs through close contact between cells. A previous study showed that ARB detected in MPs samples were 100–5000 times higher than those in water samples, and most multi-antimicrobial resistance bacteria came from MPs, where the detection rate of ARGs was 14.7% higher than that in water [129]. The enrichment ability of ARGs by different types and particle sizes of MPs is different. Among the five kinds of MPs (polyethylene, polypropylene, polystyrene, polyethylene-fiber, and polyethylene-fiber-polyethylene), the detected ARGs abundance of polyethylene and polypropylene was much higher than that of the other three kinds [130]. Similarly, exposure to millimeter-sized MPs (polyamide and polyethylene terephthalate) was found to increase the abundance of ARGs by 42.8% and 39.4%, respectively, both higher than that of microplastics at micron sizes [131]. Meanwhile, as a carrier, MPs can absorb other pollutants, such as antimicrobials and heavy metals, which can promote the prevalence of antimicrobial resistance and easily form superbugs by exerting environmental selection pressure together [128,132]. In addition, biofilm-developed MPs can absorb more pollutants and act as carriers to introduce enriched pollutants and microorganisms into organisms, thus promoting the spread of antimicrobial resistance [133].

#### 3.1.3. Other Factors

Polyaromatic hydrocarbons are another factor affecting the spread of antimicrobial resistance. A high abundance of tetracycline, sulfonamides, aminoglycosides, ampicillin, and fluoroquinolones associated with ARGs were detected in soils contaminated with polyaromatic hydrocarbons, and MGEs and integrins increased the incidence of HGT [134]. In addition, herbicides, for example glyphosate, dicamba, and glyphosate, were found to expand the abundance of ARGs and MGEs by increasing the frequency of mutations without altering the composition and diversity of soil microbial communities [135]. Fungicides are widely used pesticides, which can not only kill fungi, but also change the composition of the gut microbiome of soil animals (*Enchytraeus crypticus*) and enrich ARGs [136]. These findings highlight the important role of the environment, particularly environmental pollutants, in the spread of antimicrobial resistance, helping to create superbugs or multiple antimicrobial-resistant bacteria that threaten human health.

In addition to these environmental contaminants, some of the materials we commonly used for water treatment and soil remediation also have an impact on the spread of antimicrobial resistance. Chlorine disinfection is a way of treating drinking water and sewage. However, chlorine disinfection releases ARGs and MGEs into the environment after killing ARB, where they are ingested by chlorine-resistant microorganisms through natural genetic transformation, thereby facilitating the spread of ARGs [137]. Biochar is a good choice for soil remediation, but humic-acid-like substances in their dissolved components have been found to drive the transfer of bacterial ARGs [138]. In addition, nanomaterials are considered to be one of the most successful emerging technologies to treat soil and water pollutants, but some studies have found that silver nanoparticles promote the transfer of ARGs [139]. Therefore, the application of these technologies to address pollutants and reduce antimicrobial resistance should be considered more comprehensively.

### 3.2. The Potential Threat of Antimicrobial Resistance to Humans and the Ecological System

Pathogens acquire resistance factors from the environment and are less sensitive to antimicrobials, thereby considerably weakening our ability to prevent and cure microbial infections (Figure 2). Compared with ARGs, ARB have the most direct impact on humans and ecology. A study indicated that livestock farms are the source of large amounts of ARB and ARGs, which contaminate the surrounding environment, and have been detected antimicrobial-resistant human pathogenic bacteria (ARHPB) in the surrounding soil and water [140]. This may send a dangerous signal, but people have no effective way to limit the spread of antimicrobial resistance. Wastewater reuse is an important strategy to save freshwater resources, but WWTPs commonly fail to reach the anticipated effect on the treatment of ARB and ARGs. Treated wastewater is used for agricultural irrigation and has been demonstrated to be beneficial for the diffusion of ARB and ARGs [141]. Dispersal of ARB and ARGs into the environment may disrupt microbial community diversity [142].

ARGs in endophytic bacteria may be potential threats that have been often ignored. A recent study has shown that ARGs carried by *Escherichia coli* can be picked up by the plant microbiome in the presence of soil bacteria [143]. At the same time, a large number of studies have discovered that endophytic bacteria not only overlap with wastewater bacterial communities, but also with intestinal microbial communities [144]. Therefore, we speculated that bacteria carrying resistance factors in the environment colonized plants during the growth process through water irrigation and composting. While humans eat these unprocessed fruits and vegetables, such as strawberries, cucumbers, and radishes, endophytic bacteria can enter the gut and colonize. Ultimately, these antimicrobial-resistant bacteria enter the environment through excreta, forming a cycle between humans and the environment. However, there is currently no detailed insight into the human health assessment of endophytes carrying ARGs.

The threat of ARB to human health is obvious. ARHPB or multidrug-resistant bacteria in the environment can cause infection through the respiratory tract, esophagus, and skin contact [116,145]. Common bacterial infections cause many adverse reactions in humans, and pathogens carrying resistance genes increase the difficulty of treatment (Table 2). In addition, gut microbiota have been shown to play an important role in human health and disease, and the gut is a major habitat for ARB and ARGs [146]. It has been reported that ARGs can perform HGT in the gut [147]. All that matters is that infants show a higher abundance of ARGs in the gut than adults, and many ARGs and MGEs are of maternal origin, which indicates that ARGs or ARB can enter into the newborn via mother-to-children transmission [148]. This can be detrimental to the healthy gut microbiome composition of newborns, especially preterm infants who receive antimicrobials, who are prone to diarrhea and metabolic diseases [149]. Therefore, the development of a systematic human and environmental health risk assessment for antimicrobial resistance is essential.

## 4. Methods to Mitigate Antimicrobial Contamination and Antimicrobial Resistance

In response to the global problem of antimicrobial contamination, degrading antimicrobials in the environment is one of the most important steps. At the same time, given the global health threat of antimicrobial resistance, there is a need to understand the effectiveness of existing removal strategies and develop new treatments for ARGs and ARB. At present, the removal of antimicrobials, ARB, and ARGs mainly occurs in WWTPs. However, conventional WWTPs can significantly reduce bacterial load, but the contribution to abating ARGs is limited [157,158,159]. In addition, studies have found that ingesting ARGs from the environment through HGT is a possibly easier option than evolving and mutating to obtain resistance under antimicrobial pressure [145]. Therefore, the mitigation and treatment of antimicrobial resistance and antimicrobial contamination from the perspective of removing ARGs and antimicrobials may be a more effective strategy. Constructed wetlands (CW), advanced oxidation processes (AOP), membrane processes, and microbial fuel cells (MFC) have been reported as promising elimination methods for pollutants [122,160,161,162,163]. In the following, we summarize the existing, relatively efficient and promising antimicrobial and ARGs elimination methods.

### 4.1. AOP

AOP technologies, such as Fenton, photocatalysis, activated pyruvate, etc., can degrade harmful substances in water by strong oxidants [164]. As an organic pollutant, antimicrobials have shown good degradation effect after AOP treatment. Different oxidation mechanisms lead to different antimicrobial breakdown pathways [165]. However, AOP alone has limited effectiveness in removing complex contaminants. Therefore, AOP and other techniques can be used in combination to achieve better removal; for example, membrane treatment technology, photolysis, and biological treatment methods [166,167]. A study reported that under optimal conditions, the maximum degradation efficiencies of sulfasalazine, sulfamethoxazole, sulfamethazine, and metronidazole were 98.10%, 89.34%, 86.29%, and 58.70% respectively by O_3_/H_2_O_2_, which were much higher than that of single ozone treatment [168].

In addition, AOP technologies have also shown great performance in removing ARGs. Fenton reaction mainly produces hydroxyl radicals (·OH) with strong oxidation ability through Fe^2+^ and H_2_O_2_ reaction, thereby showing good performance in the elimination of ARB and ARGs [169]. Luo et al. optimized the Fenton process by single factor and response surface experiments, and the removal rate of ARGs after optimization was increased to 10.91~66.86% and 48.02~76.36%, respectively [169]. Compared with other AOP techniques, persulfate oxidation has a wide pH range and selectivity, and may be a future direction for exploration [165]. The photo-Fenton reaction is an optimized process that combines light, including UV, sunlight, and visible light. According to the report, nearly 60% of ARGs associated with tetracycline, sulfonamides, and macrolides were removed by solar-Fenton at neutral pH, with the complete removal of ARGs related to β-lactams and fluoroquinolones [170]. In fact, some studies have reported that although the photo-Fenton reaction has an excellent removal effect on free ARGs, it has a limited removal effect on intracellular ARGs [171]. In order to remove intracellular ARGs, increasing the concentration to produce more free radicals may be needed, but a large number of residual free radicals may cause harm to organisms. Therefore, the pretreatment of sewage to kill bacteria is a more sensible choice. Tang et al. conducted membrane filtration treatment of flowing wastewater containing antimicrobials by peroxydisulfate coupling photocatalysis, which not only achieved good degradation of tetracycline, but also reduced the emergence and transmission risk of ARGs [172].

Although AOP technologies have good effects in removing pollutants, the disadvantages are also obvious. The operational and maintenance costs have to be considered, which is crucial to practical application. It is difficult for a single AOP to mineralize antimicrobials and completely remove ARGs, so it is necessary to understand the toxicity of intermediate products in the degradation process. In addition, in order to deal with antimicrobial resistance, how to improve the ability of AOP to remove ARGs is a key point. Meanwhile, the application direction of AOP is mainly for wastewater, and its use for the removal of antimicrobials and resistant genes in soil and feces is limited.

### 4.2. CW

CW have already been used in wastewater treatment because of their low cost, and can be effective in removing organic pollutants [173]. A previous study elucidated that CW had certain removal effects on antimicrobials, ARB, and ARGs [174]. Substrate adsorption, microbial degradation, and macroplant absorption and degradation are the main mechanisms of pollutants removal in CW [175]. However, the removal effect of different CW was affected by constructed wetland type, filler type, ambient temperature, and plant type [176]. The researchers used the CW hybrid system to treat domestic wastewater and achieved a 93% removal of ciprofloxacin [177]. Another study showed that the antimicrobial removal rate of domestic sewage treated with mixed artificial humidity was 28.5–99.4%, and the combination of *Ipomoea aquatica forssk* was more effective for antimicrobial removal [178]. Full scale CW showed antimicrobial removal from 13% to 100%, and Log removal of ARGs in the water phase was low (0.8 to 1.5 log) [179]. Importantly, the CW removal rate of ARGs in the vertical flow configuration was greater than 50%, and that in the horizontal flow configuration was greater than 60% [162,180]. Avila et al. found that the use of vertical flow CW for urban wastewater treatment could effectively remove antimicrobials, and the removal rate of ARGs was between 21% and 93% [181]. Although ARGs may be less detected in outflow water, these pollutants are inevitably enriched in CW. As a result, CW needs a long time for decomposition processing. Not only that, but the removal of ARGs in CW often shows a high degree of variability over time [181].

CW are not as effective as AOP as a way to remove pollutants, but they are considered an ideal green technology due to their low construction and maintenance costs and sustainability. It has been reported that intermittent aeration was beneficial to improve water quality and remove pollutants in CW. In view of the removal effect of CW, it may be a better application direction to use it as secondary treatment of municipal wastewater after WWTPs treatment in the future. It makes sense to explore the combined effects of CW and other technologies, as well as the residual and decomposition of pollutants in CW.

### 4.3. MFC

MFC is another new type of promising green biotechnology, which can degrade pollutants and has economic benefits. After treating swine wastewater containing sulfonamides in MFC, the removal efficiencies of sulfamethoxazole, sulfadiazine, and sulfamethazine were 99.46–99.53%, 13.39–66.91%, and 32.84–67.21%, respectively [182]. In a recent study, Long et al. developed the air cathode microbial fuel cell (MFC) and MFC-Fenton systems to degrade tetracycline [183]. In MFC-Fenton cathodes, more than 99% are removed within 8 h at tetracycline concentrations of 10–40 mg/L. External resistance (R_ext_) and current intensity play a critical role in the performance of MFC in antimicrobial wastewater treatment and ARGs reduction [184,185]. At 1000 Ω R_ext_, sulfamethoxazole is degraded more efficiently and fewer ARGs are produced [184]. In terms of alleviating antimicrobial resistance, the number of MGEs in livestock and poultry wastewater treated with MFC was significantly reduced, and the relative abundance was reduced by 75% [186].

In addition, the combination of MFC and CW (CW-MFC) has recently been developed as a promising wastewater treatment technology. The methane emission of CW was greatly reduced, and the sulfonazine (>80%) and ciprofloxacin (>90%) were effectively removed [187]. The CW-MFC with new iron-carbon fillers is a further improvement on CW-MFC, which is more conducive to microbial adhesion and improves the efficiency of micro-electrolysis reaction. The efficiency of the removal of ciprofloxacin was also significantly higher than CW-MFC [188]. Notably, MFC has also been applied to the in situ remediation of antimicrobial-contaminated soils, which reduced antimicrobial residues in the soil and the risk of the release of ARGs into the soil environment [189].

### 4.4. Other Technologies

Microalgae treatment technology has been widely reported to be applied in wastewater treatment, which mainly removes antimicrobials in water through adsorption, indirect photodegradation, accumulation, and hydrolysis [166]. Its combination with bacteria, UV, or activated sludge is a popular choice to improve the antimicrobial removal rate. At present, microalgal-based treatment techniques have been shown to have a removal rate of more than 90% for tetracycline, amoxicillin, cephalosporins, and florfenicol [190,191,192,193]. In addition, the study of the enzyme degradation of antimicrobials has aroused wide interest from researchers. Lignin peroxidase, manganese peroxidase, soybean peroxidase, horseradish peroxidase, laccase, cytochrome P450 enzyme, and β -lactamase have all shown good degradation ability to various antimicrobials, but are difficult to apply at an industrial scale for the time being [6,125,194]. Most of the above methods are aimed at the elimination of antimicrobials in sewage. Composting is a biological remediation technology to remove veterinary antimicrobials in feces, although the removal effect may be affected by the type of antimicrobials [195]. Lin et al. revealed that the addition of activated carbon composites reduced antimicrobials in compost by 4–64% [196]. A recent study indicated that the intercropping of different plants may have a positive effect on antimicrobial removal [197].

A membrane bioreactor, which consists of membrane filtration and a bioreactor, is a potential wastewater treatment method [198]. Le et al. used a membrane bioreactor to effectively remove antimicrobials, ARB, and ARGs in municipal wastewater, and found that the membrane bioreactor was superior to conventional activated sludge in the elimination of ARGs [199]. In another study, a 99.79% removal rate of ARGs in swine wastewater was achieved by ultrafiltration and two-stage reverse osmosis integrated membrane filtration [161]. The efficient removal of ARGs by membrane processes can greatly reduce the occurrence and propagation of antimicrobial resistance in the environment. However, as with CW, this technology does not mean the thorough disappearance of ARGs. Although there may be some enzymes that can decompose ARGs, a large amount of ARGs have persisted on activated sludge or membranes due to the limit of environmental conditions and quantities. Further processing to achieve the desired requirements requires more data support.

The existing methods showed the certain removal ability of antimicrobials and ARGs, alleviating antimicrobial contamination and spread of antimicrobial resistance to a certain extent. Nevertheless, due to the coexistence of numerous pollutants in the environment rather than a single antimicrobial or ARGs, competing removal leads to the possibility that, in reality, these treatment technologies may not achieve the desired results. At the same time, WWTPs, for example, often consider the balance between the cost and the removal efficiency of diverse pollutants, and the temperature and pH used may not be the best removal conditions for antimicrobials and ARGs. With the ongoing concern regarding antimicrobial contamination and antimicrobial resistance, the establishment of specialized treatment facilities in hospitals and aquaculture sites, which have high concentrations of antimicrobials, ARB, and ARGs, is an effective means by which to reduce the occurrence and spread of antimicrobial resistance in the process of wastewater pipeline transportation and WWTPs.

In addition to weakening antimicrobial resistance through numerous technologies by eliminating ARGs and ARB from the environment, attention has also been focused on the discovery of alternatives to antimicrobials and adjuvant therapy means to deal with antimicrobial-resistant pathogens (Table 3). These are the main directions for future research, but achieving human clinical application remains a formidable challenge.

## 5. Discussion and Prospects

In summary, the spread of antimicrobials and antimicrobial resistance is threatening human health and the ecological environment, and the phenomenon and crisis may be greater than is generally considered. Under the interference of human factors, antimicrobials, ARB, and ARGs have been gradually surrounding human beings and filling the whole natural environment in the past century. At present, according to existing studies, antimicrobial pollution has more serious side effects on the ecological environment, while antimicrobial resistance presents greater potential harm to human health. Notably, various environmental factors, especially some pollutants such as heavy metals and microplastics, play a non-negligible role in the occurrence and spread of antimicrobial pollution and antimicrobial resistance. Due to the complexity of the environment, the combined effects of many factors are unknown and variable. In addition, an important knowledge gap is that the proliferation and prevalence of ARB and ARGs in the environment cannot be quantified, and laboratory studies have largely focused on the impact of a single factor or several factors on the spread of antimicrobial resistance. At the same time, the lack of rapid and sensitive ARB and ARGs detection means that real-time detection of sewage discharge cannot be achieved. Therefore, it is difficult to develop uniform and accepted criteria for assessing the spread and risk of antimicrobial resistance. Controlling the use of antimicrobials at source remains a major problem. Although regulations and strategies on antimicrobial use have been actively implemented in countries over the past decade, little progress has been made, especially in developing countries. Lowering the unnecessary use of antimicrobials and achieving therapeutic effects through the combination of multiple antimicrobials are commonly used in clinical practice. Nevertheless, due to the lack of relevant laws and regulations for veterinary and agricultural antimicrobials, the flux of antimicrobials into the environment is still huge.

How to better remove antimicrobials and ARGs is the key to mitigating antimicrobial pollution and the development of antimicrobial resistance. At present, the vast majority of the removal process occurs in WWTPs. However, due to the competitive removal of pollutants, unified and centralized wastewater treatment may not be sufficient to meet the requirements for antimicrobial resistance removal. The combination of multiple technologies is a good choice, but it is necessary to consider how to arrange and combine these technologies to maximize the efficiency of pollutant and antimicrobial resistance removal. At the same time, as two of the most important sources of pollution, the diffusion and spread of antimicrobial resistance in the process of pipeline transportation of hospital and aquaculture wastewater are more likely to occur, which virtually increases the burden of WWTPs. Therefore, it is necessary to set up treatment equipment for antimicrobials, ARB, and ARGs in hospitals and livestock farms to achieve initial removal. AOP has shown good performance in degrading antimicrobials and removing ARGs, which can be used as the main technology for wastewater treatment in hospitals and large farms.

In order to better deal with antimicrobial contamination and the spread of resistance, future research directions will focus on, but are not limited to, the following points:(1)To explore the possible human health risks caused by residual antimicrobials and ARGs in food, which is a topic that has been neglected and deserves attention;(2)Make an accurate and systematic risk assessment of antimicrobial and antimicrobial resistance based on environmental factors;(3)Develop fast and real-time monitoring means for antimicrobials and ARGs, and establish uniform standards for the effluent discharge of antimicrobials and ARGs;(4)Most of the research on the removal methods of antimicrobials and ARGs has been focused on the laboratory scale, which requires the establishment of practical application test sites. Data from the test site is often necessary and reliable;(5)To develop broad-spectrum, efficient, and safe antimicrobials and ARGs removal technologies is also indispensable.

## Figures and Tables

**Figure 1 toxics-11-00185-f001:**
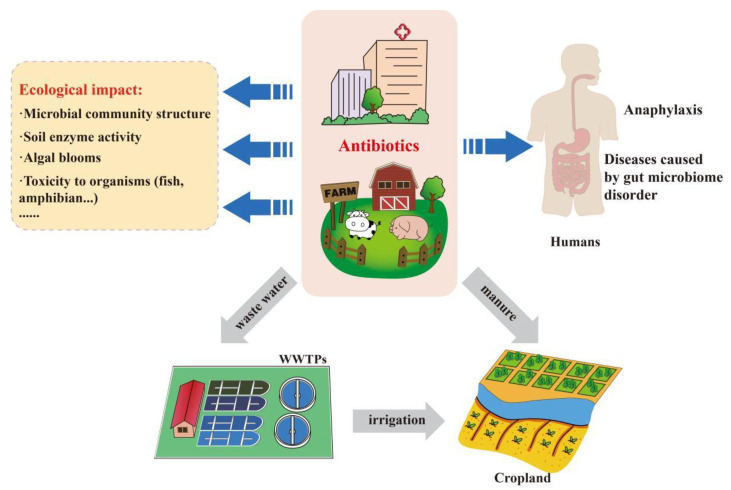
Environmental transmission of antimicrobials and impact on human health and ecology. Antimicrobials are mainly used in hospitals and farms, and some of them are removed by WWTPs, while the rest are spread into the environment. Antimicrobials entered into the environment have adverse effects on ecology and affect human health.

**Figure 2 toxics-11-00185-f002:**
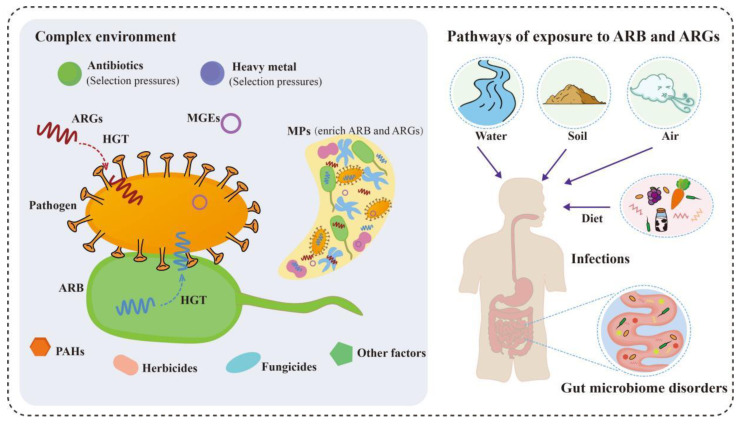
The impact of environmental pollutants on the spread of antimicrobial resistance and the pathways of human exposure to ARB and ARGs. Pollutants in complex environments promote the spread of ARGs, the reproduction of ARB, and the generation of the drug resistance of pathogens. The pathways of human exposure to ARB and ARGs in the environment mainly include skin contact, respiratory inhalation, and dietary intake, resulting in infections and gut microbiome disorders.

**Table 1 toxics-11-00185-t001:** The effects of antimicrobials on organisms in an experiment.

Antimicrobials	Organism	Antimicrobial Concentration	Effects	Reference
A five-component mixture (amoxicillin, ciprofloxacin, spiramycin, sulfamethoxazole, and tetracycline)	*Microcystis aeruginosa* NIES-843	50–500 ng/L	increased the concentration of microcystin, promoted the growth	[56]
Erythromycin	*Microcystis flos-aquae*	>10 μg/L	inhibited the growth and photosynthesis	[60]
Tilmicosin	Danio rerio	>2 mg/L	induced deformities and lethality	[63]
Oxytetracycline	*Danio rerio*	1 μg/L, 100 μg/L	affected the thyroid hormone homeostasis, reduced tryptophan hydroxylase	[64]
Norfloxacin	Carp	100 ng/L	induced the oxidative stress, damaged the barrier function of the intestine	[65]
Amoxicillin	BALB/c mice	50 mg/kg/day	reduced recognition memory, increased depression	[68]
Enrofloxacin and ciprofloxacin	*Rhinella arenarum*	>10 μg/L	affected the development, growth	[69]
Chlortetracycline	*Brassica campestris*	1 mg/L	shortened primary root length, decreased chlorophyll level	[72]
Enrofloxacin	Soybean	10 μg/L	reduced yield	[74]

**Table 2 toxics-11-00185-t002:** ARGs carried on common pathogens and related diseases.

Pathogens	Diseases	Treatment Antimicrobials	ARGs	Reference
*Staphylococcus aureus*	endocarditis, pneumonia	Lincomycin, Vancomycin	*mecA*, *VanA*	[150]
*Streptococcus pneumoniae*	pneumonia, bacteremia	Amoxicillin, Piperacillin	*Mef*, *emr(B)*	[151]
*Enterococcus*	urinary tract infection, endocarditis	Ampicillin, Piperacillin	*VanA*, *ermB*, *tet(L)*, *cat, parC*, *Cfr*	[152]
*Klebsiella pneumoniae*	pneumonia, pulmonary abscess, endocarditis	Gentamicin, Kanamycin	*bla*_KPC-2_, *bla*_CTX-M-14_, *bla*_TEM-1_, *dfrA25*	[153]
*Pseudomonas aeruginosa*	cystic fibrosis, ventilator-associated pneumonia	Gentamicin, Meropenem, Ceftazidime	*crc*, *lon*, *psrA*, *ampD*, *gyrA*, *nalA*, *nfxB*, *cbrA*	[154]
*Acinetobacter baumannii*	respiratory infections, urinary tract infection, meningitis	Imipenem, Polymyxin B	*pmrC*, *bla*_ADC-25_, *aadA*, *macA*, *gyrA*, *oprD*, *rpoB*	[155]
*Escherichia coli*	gastrointestinal infection, urinary tract infection, arthritis, meningitis	Ampicillin, Amoxicillin	*cmlA*, *flor*, *aadA*, *sul1*, *sul2*, *tetA*, *bla*_CTX-M_, *bla*_TEM_, *aphA3*, *qnrA*, *qnrS*, *OqxAB*	[156]

**Table 3 toxics-11-00185-t003:** New research related to alleviating antimicrobial resistance.

Medicine	Object of Study	Effects	Application	Reference
*Macleaya cordata* extract	*Enchytraeus crypticus*	Decreases ARGs abundance	Anti-inflammatory effects, as a feed supplement	[200]
Melatonin	*Escherichia coli* DH5α	Prevents plasmid-mediated binding transfer of antimicrobial resistance genes by disrupting proton dynamics	As inhibitors of ARGs transmission	[201]
Biotin biosynthesis inhibitors (MAC13772)	*Escherichia coli*, *Klebsiella pneumoniae*, *Pseudomonas aeruginosa*, *Staphylococcus aureus*, *Acinetobacter baumannii*	Inhibits biotin synthesis to kill the pathogenic bacterium	Infection treatment	[202]
RecA Inhibitors	*Escherichia coli* ATCC25922	Enhance the activity of bactericidal antimicrobials, reduce the acquisition of antimicrobial resistance mutations, block the horizontal transfer of mobile genetic elements	As an adjunct to antimicrobials	[203]
Glutamine	multidrug resistant *Escherichia coli*	Promotes bacterial uptake of antimicrobials to kill multidrug-resistant uropathogenic bacteria	As an adjunct to antimicrobials	[204]
Teixobactin	*Staphylococcus aureus*, *Mycobacterium tuberculosis*, *Eleftheria terrae*	Inhibits cell wall synthesis to kill pathogens without detectable resistance	A new antimicrobial	[205]

## Data Availability

Not applicable.

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
