# Peer review of "Antimicrobial and the Resistances in the Environment: Ecological and Health Risks, Influencing Factors, and Mitigation Strategies"

_toxics, 2023, doi:10.3390/toxics11020185_

Round 1
Reviewer 1 Report
This work shows an important aspect of antimicrobial overuse and its environmental consequences. I found the information relevant, appropriate for the field, and well documented.
Reviewer 2 Report
The work is interesting, but the authors tell stories rather than present scientific facts. Accordingly, the work needs to be further refined and supported with scientific facts. The Figures are not at the level of a scientific journal, they need to be corrected/replaced with new ones. The comments are included in the paper.

Reviewer 3 Report
The manuscript 'Antibiotics and antibiotic resistance: ecological and health
risks, environmental factors, and mitigation strategies' provides a comprehensive overview of enviromental implications of wide antibiotic use and resistance emergence. The manuscript is well organized, logically presented, well written, and therefore of interest to readers of the Toxics.
There are several minor issues:
1. The information on the funding is provided in the Acknowlegment section instead of the Funding section.
2. The Introduction section would benefit from the review of other review article on the related topics. Now there are no clear inclusion criteria for the cited experimental articles (e.g. time period).
Reviewer 4 Report
Comments to the authors for evaluating the following Review
Title:
Antibiotics and antibiotic resistance: ecological and health risks, environmental factors, and mitigation strategies
· Title; the first part of the tittle need rephrasing it isn’t recommended to repeat the word “ antibiotics” in addition “antibiotic resistance” must be pleural. Additionally the title did not reflect the aim of this review
· Keywords: Key words of the manuscript are too long and uninformative, you must concise them.
· The end of the abstract must provide the overall practical implementation of your results or other hypothesis that may be utilized in the future.
· The introduction part failed to emphasize “antimicrobial resistance crises worldwide” more information and recent studies must be added
· In figure 1 and 2 what is the meaning of gut microbiome disorders?
· I noted that all discussion generally on the antibiotics without any example. You have to add examples such as “low bioavailability, medical and veterinary antibiotics are often excreted by their hosts in the form of feces and urine after use” of course not all antibiotics you must explore some example
· I am not convinced about this conclusion in line 269 “Therefore, heavy metals may also act as a selective pressure for antibiotic resistance, promoting bacterial acquisition of MGEs and ARGs” as you talk over the increasing in the expression of resistance gene upon exposed to heavy metal therefore the acquisition of resistance genes aren’t correlated with the heavy metal contamination. Please provide more detail
· The word antibiotics in this review isnot suitable as you discuss some of synthetic drugs such as sulfonamides therefore, you have to replace the “antibiotic” word by “antimicrobial”
· This review failed to discuss the detail information and the association between the antimicrobial contamination and the spreading of resistance
· Generally, this review superficially discuss the environmental factors that can exacerbate antibiotic contamination and the spread of antibiotic resistance and there is no accuracy in the interpretation of any point in this review
Round 2
Reviewer 2 Report
The paper can be accepted in present form.
Reviewer 4 Report
thank you for providing the new version of this manuscript
all my recommendation were well managed but the tittle need more modifications